# FLAIR: Storing Unbounded Data Streams on Mobile Devices to Unlock User Privacy at the Edge

## Abstract

Mobile devices are producing larger and larger data streams, such as location streams, which are consumed by machine learning pipelines deliver location-based services to end users. Such data streams are generally uploaded and centralized to be processed by third parties, potentially exposing sensitive personal information. In this context, existing protection mechanisms, such as *Location Privacy Protection Mechanisms* (*LPPMs*), have been investigated. Alas, none of them have actually been implemented, nor deployed in real-life, in mobile devices to enforce user privacy at the edge. We believe that the effective deployment of LPPMs on mobile devices faces a major challenge: the storage of unbounded data streams. This paper introduces FLAIR, a storage system based on a new piece-wise linear approximation technique that increases the storage capacity of mobile devices by relying on data modeling. Beyond the FLAIR storage layer, we also introduce *Divide & Stay*, a new privacy-preserving technique to execute *Points of Interest* (POIs) inference. Finally, we deploy both of them on Android and iOS to demonstrate that a real deployment of LPPMs is now possible.

## 1 Introduction

With the advent of smartphones and more generally the *Internet of Things* (IoT), connected devices are mainstream in our societies and widely deployed at the edge. Such constrained devices are not only consuming data and services, such as streaming, geolocalization, or restaurant recommendations, but also producers of data streams by leveraging a wide variety of embedded sensors that capture the surrounding environment of end-users, including their daily routines. Online services are heavily relying on this crowdsourced data to improve the user experience through machine learning. The data deluge generated by a user is potentially tremendous: according to preliminary experiments, a smartphone can generate approximately 2 pairs of GPS samples and 476 triplets of accelerometer samples per second, resulting in more than 172,800 location and 41,126,400 acceleration daily samples. These data streams tend to be uploaded from the device to third-party service providers to extract the valuable information it contains. As an example, the *Point Of Interest*s (POIs) of a user can be extracted from her GPS traces, to better understand consumers' behavior.

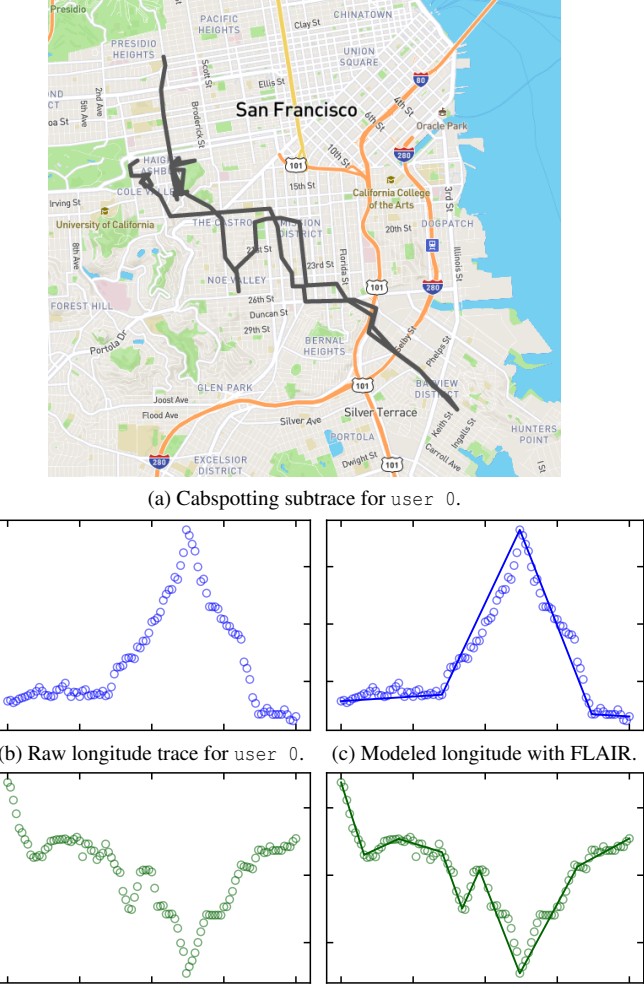

(a) Cabspotting subtrace for `user 0`.

(b) Raw longitude trace for `user 0`.  (c) Modeled longitude with FLAIR.

(d) Raw latitude trace for `user 0`.  (e) Modeled latitude with FLAIR.

Figure 1: FLAIR compacts any location stream as a sequence of segments, obtained from a piece-wise model.

However, this continuous data stream inevitably includes *sensitive personal information* (SPI) that may jeopardize the privacy of end-users, if processed by malicious stakeholders. While machine learning algorithms are nowadays widely adopted as a convenient keystone to process large datasets and infer actionable insights, they often require grouping raw input datasets in a remote place, thus imposing a privacy threat for end-users sharing their data. This highlights the utility vs. privacy trade-off that is inherent to any data-sharing activity. On the one hand, without crowd-sourced GPS traces, it would be hard to model traffic in real-time and recommend itineraries. On the other hand, it is crucial to protect user privacy when accepting to gather SPI.

To address this ethical challenge, privacy-preserving machine learning [39] and decentralized machine learning [16, 40] are revisiting state-of-the-art machine learning algorithms to enforce user privacy, among other properties. Furthermore, regarding location privacy, several protection mechanisms, called *Location Privacy Protection Mechanism*s (LPPMs), have been developed to preserve user privacy in mobility situations. Location reports are evaluated and obfuscated before being sent to a service provider, hence keeping user data privacy under control. The user no longer automatically shares her data streams with service providers, but carefully selects what she shares and makes sure the data she unveils does not contain any SPI. For example, Geo-Indistinguishability [15] generalizes differential privacy [18] to GPS traces, while PROMESSE [35] smooths the GPS traces—both temporally and geographically—to erase POIs from the input trace. LPPMs successfully preserve sensitive data, such as POIs, while maintaining the data utility for the targeted service.

Despite their effectiveness, no LPPM has ever been implemented and deployed on mobile devices: previous works have been simulated on ADB [25] at best. While extending those works to Android and iOS devices may be perceived as straightforward, it faces several challenges imposed by the scarce resources of mobile devices. In particular, LPPMs often require the user to access all her GPS traces and, ideally, the ones of additional users. The strategy consisting in storing entire raw traces does not scale and is impracticable for the average user who does not possess high memory devices. Unfortunately, the memory constraints of modern devices prevent the users from sharing user traces at the edge of the network.

This paper demonstrates that modeling data streams make this transfer possible. In particular, we introduce *Fast LineAr InteRpolation* (FLAIR), a new data storage system based on a new piecewise linear approximation technique, and we use it to model and store data streams under memory constraints (see Fig. 1). Unlike existing stream or temporal databases, FLAIR does not store a fixed number of data samples but models their evolution, theoretically offering an unlimited storage capacity. We show that FLAIR can be deployed on Android and iOS smartphones to store GPS traces of entire datasets. We then implement a LPPM working directly on mobile phones—which is made possible by the increased GPS storage capacity offered by FLAIR. However, the LPPM's privacy gains need to be evaluated *in situ* before being uploaded to service providers: are POIs actually obfuscated? To this end, we also introduce a new POI attack algorithm, dubbed *Divide & Stay* (D&S), which can compute POI on large traces in tens of seconds directly on mobile phones. We report that our combined approaches enable storing tremendous amounts of geolocation data on mobiles, thus allowing the use of LPPMs to ensure end-user privacy while using geolocation services.

In the following, we first discuss the related work (Sec. 2), before diving into the details of FLAIR and how it can be applied to boost location privacy (Sec. 3). We then present our experimental setup (Sec. 4) and the results we obtained (Sec. 5); we discuss the potential shortcoming of our approach (Sec. 6) before concluding (Sec. 7).

## 2   Related Works

### 2.1   Location Privacy Attacks

Raw user mobility traces can be exploited to model the users' behavior and reveal their *sensitive personal information* (SPI). In particular, the *Point Of Interest*s (POIs) are widely used as a way to extract SPI from mobility traces. In a nutshell, a POI is a place where the user comes often and stays for a significant amount of time: it can reveal her home, workplace, or leisure habits. From POIs, more subtle information can also be inferred: sexual orientation from attendance to LGBT+ places, for instance. The set of POIs can also be used as a way to re-identify a user in a dataset of mobility traces [21, 34]. The POIs can be extracted using spatiotemporal clustering algorithms [23, 42]. Alternatively, an attacker may also re-identify a user directly from raw traces, without computing any POI [29].

### 2.2   Mobility Dataset Protection Mechanisms

When data samples are gathered in a remote server, one can expect the latter to protect the dataset as a whole. In particular, *k-anonymity* [36] is the property of a dataset guaranteeing that whenever some data leaks, the owner of each data trace is indistinguishable from at least $k-1$ other users contributing to the dataset. Similarly, *l-diversity* [27] extends *k-anonymity* by ensuring that the $l$ users are diverse enough not to infer SPI about the data owner. Finally, *differential privacy* [18] aims at ensuring that the inclusion of a single element in a dataset does not alter significantly an aggregated query on the whole dataset. However, all these techniques require personal samples to be grouped to enforce user privacy.

## 2.3 Location Privacy Protection Mechanisms

Rather than protecting the dataset as a whole, each data sample can also be protected individually. In the case of location data, several protection mechanisms—called *Location Privacy Protection Mechanism*s (LPPMs)—have been developed. They may be deployed in a remote server where all data samples are gathered or directly on the device before any data exchange.

*Geo-Indistinguishability* (GEOI) [15] implements differential privacy [18] at the trace granularity. In particular, GEOI adjusts mobility traces with two-dimensional Laplacian noise, making POIs more difficult to infer. *Heat Map Confusion* (HMC) [28] aims at preventing re-identification attacks by altering all the traces altogether. The raw traces are transformed into heat maps, which are altered to look like another heat map in the dataset, and then transformed back to a GPS trace.

PROMESSE [35] smooths the mobility traces, both temporally and geographically, to erase POIs from the trace. PROMESSE ensures that, between each location sample, there is at least a given time and distance interval. In the resulting mobility trace, the user appears to have a constant speed. While PROMESSE blurs the time notion from the trace—*i.e.*, the user never appears to stay at the same place—it does not alter their spatial characteristics. Yet, while POIs may be still inferred if the user repeatedly goes to the same places, it will be harder to distinguish such POIs from more random crossing points.

It is also possible to combine several LPPMs to improve the privacy of users [25, 30]. Because of potential remote leaks, the user should anonymize her trace locally before sharing it, which is how EDEN [25] operates. However, EDEN has not been deployed: it has only been simulated on ADB. Even more so: despite their validity and to the best of our knowledge, no LPPM has been implemented in mobile devices. This is partly due to the tight constraints of mobile devices, memory-wise notably: HMC [28], for instance, requires locally loading a large set of GPS traces to operate.

## 2.4 Temporal Databases & Mobile Devices

To overcome the memory constraints of mobile devices, one needs efficient embedded temporal databases. To take the example of Android: only few databases are available, such as SQLITE and its derivative DRIFT [1], the cloud-based Firebase [3], the NOSQL HIVE, and OBJECTBOX [11]. The situation is similar on iOS.

**Relational databases** Relational databases (*e.g.*, SQL) are typically designed for *OnLine Transactional Processing* (OLTP) and *OnLine Analytical Processing* (OLAP) workloads, which widely differ from time-series workloads. In the first, reads are mostly contiguous (as opposed to the random-read tendency of OLTP); writes are most often inserts (not updates) and typically target the most recent time ranges. OLAP is designed to store big data workloads to get analytical statistics from data, while not putting the emphasis on read nor write performances. Finally, in temporal workloads, it is unlikely to process writes & reads in the same single transaction [37].

Despite these profound differences, several relational databases offer support for temporal data with industry-ready performance. As an example, TimescaleDB [14] is a middleware that exposes temporal functionalities atop a relational PostgreSQL foundation.

**InfluxDB** InfluxDB [8] is one of the most widely used temporal databases. Implemented in Go, this high-performance time series engine is designed for really fast writes to collect metrics and events from IoT sensors. Unfortunately, its retention policy prevents the storage to scale in time: the oldest samples are dumped to make room for the new ones.

To the best of our knowledge, however, none of the existing solutions prioritize data compression to the extent that they would *prune* raw data samples in favor of *modeled* approximations.

**Modeling data streams** While being discrete, the streams sampled by sensors represent inherently continuous signals. Data modeling does not only allow important memory consumption gains, but also flattens sensors' noise, and enables extrapolation between measurements. In particular, *Piecewise Linear Approximation* (PLA) are used to model the data in successive linear polynomials. An intuitive way to do linear approximation is to apply a bottom-up segmentation: each pair of consecutive points is connected by interpolations; the less significant contiguous interpolations are merged, as long as the obtained interpolations introduce no error above a given threshold. The bottom-up approach has low complexity but usually requires an offline approach to consider all the points at once. The *Sliding Window And Bottom-up* (SWAB) algorithm [24], however, is an online approach that uses a sliding window to buffer the latest samples on which a bottom-up approach is applied. emSWAB [17] improves the sliding window by adding several samples at the same time instead of one. Instead of interpolation, linear regression can also be used to model the samples reported by IoT sensors [22]. For example, GREYCAT [31] adopts polynomial regressions with higher degrees to further compress the data. Unfortunately, none of those works have been implemented on mobile devices to date.

Closer to our work, FSW [26] and the ShrinkingCone algorithm [20] attempt to maximize the length of a segment while satisfying a given error threshold, using the same property used in FLAIR. FSW is not a streaming algorithm as

it considers the dataset as a whole, and do not support insertion. The ShrinkingCone algorithm is a streaming greedy algorithm designed to approximate an index, mapping keys to positions: it only considers monotonic increasing functions and can produce disjoints segments. FLAIR models non-monotonic functions in a streaming fashion, while providing joints segments.

## 3 Enabling User Privacy at the Edge

### 3.1 *In-situ* Data Management

For privacy's sake, we advocate for *in-situ* data management strategies—*i.e.*, SPI should be anonymized within the mobile device *before* any data exchange. This avoids anonymizing by relying on a trusted third party first gathering multiple users' raw data. Such a third party may accidentally or intentionally leak users' data, making the adoption of such protection mechanisms ineffective.

In the following, we will focus on mobility traces. A mobility trace is an ordered sequence $T$ of pairs $(t, g)$ where $t$ is a timestamp and $g$ is a geolocation sample, a latitude-longitude pair for example. The trace is ordered in chronological order and we assume that reported timestamps are unique.

We believe that keeping the raw data where it is created—*i.e.*, on the mobile devices—increases user privacy. However, sharing data is required to enable location-based services, such as traffic modeling. The user should share their mobility traces *after* they have been protected using an LPPM. The first challenge is to find which LPPM to use and which related parameters are optimal. To tackle this issue, a public dataset can be used to estimate the impact of an LPPM and to pick the best option. EDEN [25] proposes a more advanced solution: federated learning is used among the participants to learn a model which can predict the best configuration without sharing any mobility trace. Nonetheless, both approaches require storing an important volume of data to successfully protect user privacy.

The strong resource constraints of mobile devices prevent the previous solutions to work in practice. In particular, mobile ecosystems lack system components to deploy efficient local storage solutions. Not only is there no advanced database readily available on mobile operating systems, but no native data modeling framework is provided either. For example, EDEN was implemented using the PYTORCH library [12], which is not available on smartphones[1]: the proposal was only simulated on a server. It is, therefore, crucial to deliver tools enabling the deployment of *state-of-the-art* techniques in mobile devices to support privacy-preserving strategies at the edge of a network.

---

[1] PyTorch allows importing and using trained models on Android and iOS, but disallows training them locally.

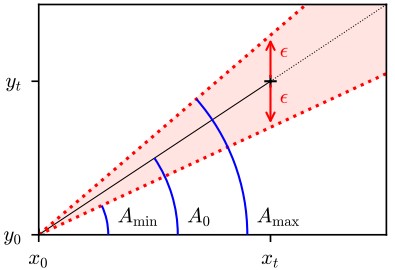

Figure 2: FLAIR considers the sample $s_0 = (x_0, y_0)$ of the current model as the origin. In addition to the current gradient $A_0$, the minimum and maximum acceptable gradients, $A_{\min}$ and $A_{\max}$, are kept. $A_{\min}$ and $A_{\max}$ are defined such that the error reported by the model is lower than or equal to $\varepsilon$. To check if a new sample $s_t = (x_t, y_t)$ fits the model, FLAIR computes its gradient $A_t$ and compares it to $A_{\min}$ and $A_{\max}$.

### 3.2 Unleashing Your Device Storage with FLAIR

To overcome the memory constraint of mobile devices, efficient temporal databases must be ported onto mobile environments. In particular, we advocate the use of data modeling, such as PLA [22, 24] or GREYCAT [31], to increase the storage capacity of constrained devices. We propose *Fast LineAr InteRpolation* (FLAIR), a storage system based on a fast PLA to store approximate models of any data stream on any mobile device, instead of storing all the raw data samples as state-of-the-art temporal databases do. For simplicity, we refer to both the storage system and the associated modeling technique as FLAIR.

FLAIR models one-dimensional samples as piece-wise linear interpolations that enforce the following invariant: *all samples modeled by an interpolation must maintain an error below the configuration parameter $\varepsilon$*. Data samples are inserted incrementally: the current model is adjusted to fit new samples until it cannot satisfy the invariant. In that case, the current model is persisted in memory $\mathcal{M}$, and a new interpolation begins from the two last inserted points. Each model in $\mathcal{M}$ is represented by a pair $(s_i, A_i)$: $s_i = (x_i, y_i)$ is the interpolation's initial sample, while $A_i$ is the line's gradient. Each model thus represents the function $y = A_i \times (x - x_i) + y_i$. While working on the current model, its initial sample is set as the origin $s_0 = (x_0, y_0)$, the current interpolation is thus a polynomial defined as $y = A_0 \times x$. The current gradient $A_0$ is the slope between $s_0$ and the last interpolated sample $s_t$. Fig. 2 depicts a FLAIR model with two initial samples $s_0$ and $s_t$. It shows the interpolation parameters $(s_0, A_0)$, and two additional gradients $A_{\min}$ and $A_{\max}$. A naive solution to maintain the invariant while updating the current model would be to memorize every sample between $s_0$ and the last sample $s_t$, to check their error against the model. Instead, FLAIR only maintains $A_{\min}$ and $A_{\max}$, which are updated at each sample

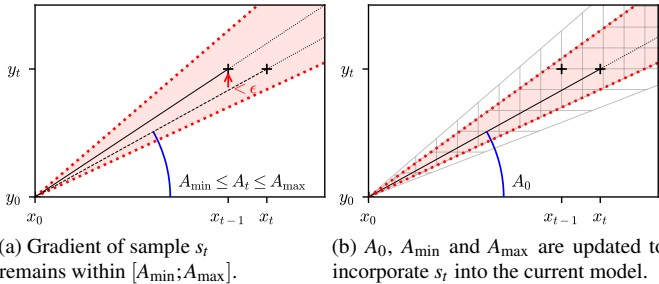

(a) Gradient of sample $s_t$ remains within $[A_{\min};A_{\max}]$.

(b) $A_0$, $A_{\min}$ and $A_{\max}$ are updated to incorporate $s_t$ into the current model.

Figure 3: When a new sample fits within $[A_{\min};A_{\max}]$, it is added to the current model by updating $A_0$ and the interval to ensure that all previous samples fit the updated model.

---

**Algorithm 1** FLAIR insertion using parameter $\varepsilon \in \mathbb{R}^{+*}$

---

**Before:** $\mathcal{M}$; $x_0, x_{t-1} \in \mathbb{R}^+$; $y_0, y_{t-1}, A_0, A_{\min}, A_{\max} \in \mathbb{R}$

1: **function** INSERT($x_t \in \mathbb{R}^+, y_t \in \mathbb{R}$)
2:     $\left(x_t^\Delta, y_t^\Delta\right) \leftarrow (x_t - x_0, y_t - y_0)$          ▷ Compute $A_t$
3:     $A_t \leftarrow y_t^\Delta / x_t^\Delta$
4:     **if** $A_{\min} \leq A_t \leq A_{\max}$ **then**
5:         $A_0 \leftarrow A_t$          ▷ Update model
6:         $A_{\min} \leftarrow \max\left(A_{\min}, \frac{y_t^\Delta - \varepsilon}{x_t^\Delta}\right)$
7:         $A_{\max} \leftarrow \min\left(A_{\max}, \frac{y_t^\Delta + \varepsilon}{x_t^\Delta}\right)$
8:     **else**
9:         $\mathcal{M}.\text{insert}(x_0, y_0, A_0)$          ▷ Persist model
10:        $(x_0, y_0) \leftarrow (x_{t-1}, y_{t-1})$          ▷ Build new model
11:        $\left(x_t^\Delta, y_t^\Delta\right) \leftarrow (x_t - x_0, y_t - y_0)$
12:        $A_0 \leftarrow y_t^\Delta / x_t^\Delta$
13:        $A_{\min} \leftarrow \left(y_t^\Delta - \varepsilon\right) / x_t^\Delta$
14:        $A_{\max} \leftarrow \left(y_t^\Delta + \varepsilon\right) / x_t^\Delta$
15:    **end if**
16:    $(x_{t-1}, y_{t-1}) \leftarrow (x_t, y_t)$          ▷ Update penultimate
17: **end function**

---

insertion.

Algorithm 1 details the insertion of a new sample $s_t$. First, FLAIR computes the gradient $A_t$ of the line $(s_0, s_t)$ (lines 2-3). If $A_t$ is inside $[A_{\min};A_{\max}]$, $s_t$ is added to the current model by updating $A_0$, $A_{\min}$ and $A_{\max}$ (lines 5-7), as displayed in figure 3. Graphically, we see that the resulting 'allowed cone' is the intersection of the model's previous one, and that of $s_t$'s allowed error. By recurrence, the cone materialized by $s_0$ and $[A_{\min};A_{\max}]$ is the intersection of the error margin of every point modeled by the current interpolation—illustrating how FLAIR respects its invariant. If $A_t$ falls outside the interval $[A_{\min};A_{\max}]$, $s_t$ breaks the invariant: the current model is persisted in memory $\mathcal{M}$ (l. 9), and a new model $(s_0, A_0)$ is computed from $s_{t-1}$, along with new limits $A_{\min}$ and $A_{\max}$ (l. 10-14). This case is displayed in figure 4. In any case, the penultimate sample $s_{t-1}$ is updated on line 16.

In FLAIR, reading a value $x$ is achieved by estimating its image using the appropriate model, as is shown in algorithm 2.

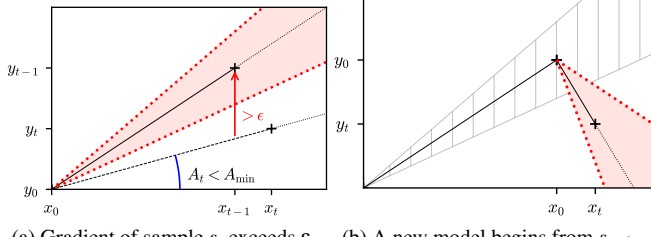

(a) Gradient of sample $s_t$ exceeds $\varepsilon$.

(b) A new model begins from $s_{t-1}$.

Figure 4: When a new sample reports an error $> \varepsilon$, a new model is created using the penultimate sample $s_{t-1}$ as $s_0$.

Lines 2-3 display the computation of the image when $x$ belongs to the current model. When it does not, FLAIR retrieves the model in charge of approximating $x$ (l. 5). In practice, this is made through a dichotomy search, as $\mathcal{M}$ stores models in insertion order. Using that model, the interpolation of $x$ is computed on line 6.

---

**Algorithm 2** FLAIR approximate read

---

**Before:** Current model $(x_0, y_0, A_0)$;
    Memory $\mathcal{M}$ containing previous models
1: **function** READ($x \in \mathbb{R}^+$)
2:     **if** $x_0 \leq x$ **then**
3:         **return** $A_0 \times (x - x_0) + y_0$
4:     **end if**
5:     Select $i$ s.t. $(x_i, y_i, A_i) \in \mathcal{M} \wedge x_i \leq x < x_{i+1}$
6:     **return** $A_i \times (x - x_i) + y_i$
7: **end function**

---

The value of $\varepsilon$ has an important impact on the performances of FLAIR. Figure 5 illustrates the longitude of Figure 1b with two extreme values for $\varepsilon$. If $\varepsilon$ is too small (Fig. 5a), none of the inserted samples fits the current model at that time, initiating a new model each time. In that case, there will be one model per sample, imposing an important memory overhead. The resulting model overfits the data. On the other hand, if $\varepsilon$ is too large (Fig. 5b), then all the inserted samples fit, and a single model is kept. While it is the best case memory-wise, the resulting model simply connects the first and last point and underfits the data.

While FLAIR is designed for the modeling of one-dimensional data, it straight-forwardly generalizes to multiple-dimensional data by combining several instances of FLAIR. As long as the newly inserted data samples fit the existing model, the memory footprint of FLAIR remains unchanged. This potentially unlimited storage capacity makes FLAIR a key asset for mobile devices, making the storage of mobility traces possible. We claim that the use of FLAIR alleviates the memory constraint of mobile devices, making the real use of LPPM possible and paving the way for user control of SPI.

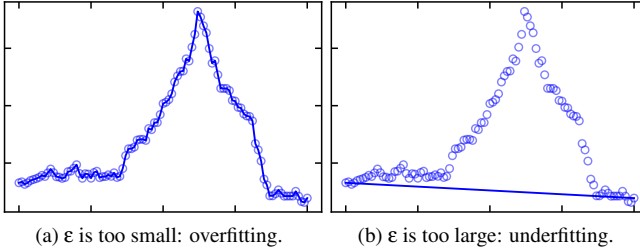

(a) ε is too small: overfitting.          (b) ε is too large: underfitting.

Figure 5: The performances of FLAIR is highly dependent on the value of ε: a too small value will result in overfitting and a too large one in underfitting.

## 3.3 Evaluating Your Location Privacy with D&S

To demonstrate that FLAIR enables the deployment of existing LPPM in the wild, we use FLAIR on a mobile device to store an entire dataset of mobility traces. Then, we perform a geolocation attack on these traces, with and without the use of an LPPM. We focus on POI attacks [34] and we use PROMESSE [35] as the LPPM to protect the mobility traces. POI-attack [34] aims at extracting the POIs from a mobility trace. The extraction is done by a two-steps algorithm: first potential candidates for POIs, dubbed *stays*, are extracted and, then, these *stays* are merged to avoid duplication of similar POIs. A *stay* is defined as a circle with a radius lower than $D_{max}$ where a user spent a time higher than a set time $t_{min}$. A *stay* is represented by its center. The two thresholds $t_{min}$ and $D_{max}$ have an important impact on the type of POI extracted. *Short stays* will identify day-to-day patterns, such as shopping preferences, while *long stays* will identify travel preferences and periods, for example. The resulting *stays* whose centroids are close to a given value are then merged to obtain the final POI.

The regular way to extract the *stays* is to iterate over the mobility traces and compute *stays* as they appear [34]. Unfortunately, this approach is very expensive for dense mobility traces—*i.e.*, with many data samples per unit of time. Instead of sampling, which results in a loss of information, we introduce a new algorithm to extract the *stays* while scaling with the density of the traces. This contribution, named *Divide & Stay* (D&S), is a divide-and-conquer algorithm that considers the mobility trace as a whole, and not iteratively.

The intuition behind *Divide & Stay* is to avoid computing *stays* when it is useless. It is impossible to extract a *stay* from a segment where more than $D_{max}$ meters have been traveled in less than $t_{min}$. For example, the mobility trace of a trip in a car at high speed in a straight line meets those conditions. While the regular approach would consider each location until the end of the trace, D&S skips it entirely. The denser the trace the more time the regular approach would spend on such segments. The key idea of *Divide & Stay* is to recursively divide the trace until either such a segment is found, and discarded, or until a fixed size segment is found on which the

regular way to extract *stays* is performed.

More precisely, in D&S, the trace is split into two parts, cut in the middle. Both segments left and right, are individually considered. If the start and endpoint of the segment are close temporally, but far spatially, it means that no *stay* would be possibly extracted: no stay is further searched on this segment. Otherwise, *stays* are recursively computed with the top-down approach on the segment, until the size is lower than a given threshold $S$, *e.g.* 300. In that case, the classical way to compute *stays* [34] is triggered on the considered subtrace. Algorithm 3 depicts the pseudo-code of *Divide & Stay*. The trace $T$ is manipulated as a whole but with the different indexes $s$, $i$, and $e$ for the recursion. $T[i].t$ refers to the timestamp of the sample $T[i]$ and $T[i].g$ refers to the associated location. The distance between two locations is computed with geo.dist and the function getStays refers to the original function computing *stays* [34].

The more discarded segments, the faster compared to the regular approach. However, *stays* around the middle points of index $i$ could be missed, but D&S ignores them as a POI is a cluster of several stays: it is very unlikely to miss them all. D&S can be implemented sequentially or concurrently, to leverage multi-core processors.

---

**Algorithm 3** *Divide & Stay* (D&S)

**Input:** $T \in (\mathbb{R} \times \mathbb{G})^n$; $S \in \mathbb{N}^+$; $s \in [\![0;n-1]\!]$;
$\quad e \in [\![0;n-1]\!], (t_{min}, D_{max}) \in \mathbb{R}^{2+}$
**Output:** $STAYS \in (\mathbb{R} \times \mathbb{G})^n$
$\quad STAYS \leftarrow \emptyset$
$\quad$**if** $T.size() \leq S$ **then**
$\quad\quad$**return** getStays($T$.subtrace($s,e$), $m,D$)
$\quad$**end if**
$\quad i = \lfloor (e+s)/2 \rfloor$
$\quad t1 = T[i].t - T[s].t$
$\quad d1 = $ geo.dist($T[s].g, T[i].g$)
$\quad$**if** $\neg(d1 > D_{max} \wedge t1 \leq t_{min})$ **then**
$\quad\quad STAYS+ = D\&S(T,S,s,i,t_{min},D_{max})$
$\quad$**end if**
$\quad t2 = T[e].t - T[i].t$
$\quad d2 = $ geo.dist($T[i].g, T[e].g$)
$\quad$**if** $\neg(d2 > D_{max} \wedge t2 \leq t_{min})$ **then**
$\quad\quad STAYS+ = D\&S(T,S,i,e,t_{min},D_{max})$
$\quad$**end if**
$\quad$**return** $STAYS$

---

## 4 Experimental Setup

This section presents observed indicators used to affirm the value of FLAIR's contribution to mobile machine learning on time series. We then introduce datasets that were used to assert FLAIR's storage capabilities. Next, we present competing solutions that were also implemented in benchmark

applications to compare with FLAIR's performances. Finally, we discuss experimentation settings.

## 4.1 Key Performance Metrics

To evaluate how our approach performs, we use two classes of key performance metrics: system metrics and privacy-related metrics. Concerning privacy-related experiments, we only measure the computation time when evaluating *Divide & Stay*. Those metrics highly depend on the chosen algorithms, while the use of FLAIR has no impact. Since our objective is to demonstrate that FLAIR can help to port state-of-the-art LPPM techniques on constrained devices, we do not discuss privacy-related metrics for other experiments.

**Memory footprint**  The key objective of FLAIR is to reduce the memory footprint required to store an unbounded stream of samples. More specifically, we explore two metrics: *(i)* the number of 64-bits variables required by the model and *(ii)* the size of the model in the device memory. To do so, we compare the size of the persistent file with the size of the vanilla SQLite database file. We consider the number of 64-bit variables as a device-agnostic estimation of the model footprint.

**I/O throughput**  Another relevant system metric is the I/O throughput of the temporal databases. In particular, we measure how many write and read operations can be performed per second.

We will be comparing POI-inference algorithms, and POIs returned by the same algorithm using different data backends. For that reason, we need two metrics to compare the sets of POIs returned in the different cases: a distance between POIs, and the sets' sizes.

Measuring the quality of inferred POIs is difficult, as there is no acknowledged definition of how to compute POIs. We consider as our ground truth the POIs inferred by the state-of-the-art POI-attack [34], which we refer to as the 'raw' POIs. The existence of such a 'ground-truth' is however debatable, as two different—but close—POIs can be merged by the algorithm into a single POI. As an example, if a user visits two different shops separated by a road, but their distance is lower than $D_{max}$, those will be merged into a single POI located at the center of the road.

**Distance between POIs**  As the POI definition is mainly algorithmic, we compute the distance of each obtained POI to its closest raw POI as the metrics assessing the quality of new POIs. These distances are reported as a *Cumulative Distribution Function* (CDF). If FLAIR does not alter significantly the locations of the mobility traces it captures, the computed distances should be short.

**Number of POIs**  In addition to the distances between POIs, we are also considering their returned quantity as a metric. In our previous example, visiting the two shops may result in two different POIs because they have been slightly shifted by FLAIR. Beyond the numbers, we expect that PROMESSE successfully anonymizes mobility traces by returning a grand total of zero POI.

## 4.2 Mobility Datasets

**Cabspotting**  CABSPOTTING [33] is a mobility dataset of 536 taxis in the San Francisco Bay Area. The data was collected during a month and is composed of 11 million records, for a total of 388MB.

**PrivaMov**  PRIVAMOV [32] is a multi-sensors mobility dataset gathered during 15 months by 100 users around the city of Lyon, France. We use the full GPS dataset, which includes 156 million records, totaling 7.2GB. Compared to CABSPOTTING, PRIVAMOV is a highly-dense mobility dataset.

## 4.3 Storage Competitors

**SQLite**  SQLITE is the state-of-the-art solution to persist and query large volumes of data on Android devices. SQLITE provides a lightweight relational database management system. SQLITE is not a temporal database, but is a convenient and standard way to store samples persistently on a mobile device. Insertions are atomic, so one may batch them to avoid one memory access per insertion.

**SWAB**  *Sliding-Window And Bottom-up* (SWAB) [24] is a linear interpolation model. As FLAIR, the samples are represented by a list of linear models. In particular, reading a sample is achieved by iteratively going through the list of models until the corresponding one is found and then used to estimate the requested value. The bottom-up approach of SWAB starts by connecting every pair of consecutive samples and then iterates by merging the less significant pair of contiguous interpolations. This process is repeated until no more pairs can be merged without introducing an error higher than ε. Contrarily to FLAIR, this bottom-up approach is an offline one, requiring all the samples to be known. SWAB extends the bottom-up approach by buffering samples in a sliding window. New samples are inserted in the sliding window and then modeled using a bottom-up approach: whenever the window is full, the oldest model is kept and the captured samples are removed from the buffer.

One could expect that the bottom-up approach delivers more accurate models than the greedy FLAIR, even resulting in a slight reduction in the number of models and faster readings. On the other hand, sample insertion is more expensive than FLAIR due to the execution of the bottom-up

approach when storing samples. Like FLAIR, SWAB ensures that reading stored samples is at most ε away from the exact values.

**Greycat** GREYCAT [31] aims at compressing even further the data by not limiting itself to linear models. GREYCAT also models the samples by a list of models, but these models are polynomials. The samples are read exactly the same way.

When inserting a sample, it first checks if it fits the model. If so, then nothing needs to be done. Otherwise, unlike FLAIR and SWAB which directly initiate a new model, GREYCAT tries to increase the degree of the polynomial to make it fit the new sample. To do so, GREYCAT first regenerates $d + 1$ samples in the interval covered by the current model, where $d$ is the degree of the current model. Then, a polynomial regression of degree $d + 1$ is computed on those points along the new one. If the resulting regression reports an error higher than $\frac{\varepsilon}{2^{d+1}}$, then the model is kept, otherwise, the process is repeated by incrementing the degree until either a fitting model is found or a maximum degree is reached. If the maximum degree is reached, the former model is stored and a new model is initiated. The resulting model is quite compact, and thus faster to read, but at the expense of an important insertion cost.

Unlike FLAIR and SWAB, there can be errors higher than ε for the inserted samples, as the errors are not computed on raw samples but on generated ones, which may not coincide. Furthermore, the use of higher-degree polynomials makes the implementation subject to overflow: to alleviate this effect, the inserted values are normalized.

## 4.4 Experimental Settings

For experiments with unidimensional data—*i.e.* memory and throughput benchmarks—we set $\varepsilon = 10^{-2}$. The random samples used in those experiments are following a uniform distribution in $[-1,000;1,000]$: it is very unlikely to have two successive samples with a difference lower than ε. For experiments on location data, and unless said otherwise, we set $\varepsilon = 10^{-3}$ for FLAIR, SWAB and Greycat. For Greycat, the maximum degree for the polynomials is set to 14. For POI computations, we use $t_{min} = 5$ min and a diameter of $D_{max} = 500$ m for both the standard approach and D&S. Similarly, we use $\delta = 500$ m for PROMESSE: it should remove all the POIs from the traces.

The experiments evaluating the throughput were evaluated four times each and the average is taken as the standard deviation was small. All the other experiments are deterministic and performed once.

## 4.5 Implementation Details

We ran our experiments on a Fairphone 3 [2] running Android 11; we reproduced them on an iPhone 12 [9] running iOS 15.1.1. We chose to implement our evaluation apps using Flutter [6]. Flutter is Google's UI toolkit, based on the Dart programming language, that can be used to develop natively compiled apps for Android, iOS, web and desktop platforms (as long as the project's dependencies implement cross-compilation to all considered platforms).

We, therefore, implemented a Flutter library including FLAIR, its storage competitors, the POI-attack with and without our D&S extension, and PROMESSE. Our implementation is publicly available [5]. For our experiments, we implemented several mobile applications based on this library.

## 5 Experimental Results

In this section, we evaluate our implementation of FLAIR on Android and iOS to show how it can enable *in-situ* data management on mobile devices. We first show that using FLAIR paves the way for storing a tremendous quantity of samples, by comparing it to SQLITE and reporting its performances when storing samples generated by the accelerometer. Then, we deploy the PROMESSE LPPM directly on mobile thanks to FLAIR. Still on the mobile phones, we evaluate traces using our POI-attack *Divide & Stay* (D&S): to assess the precision of the GPS time series modeled by FLAIR, and the privacy gain of the LPPM.

## 5.1 Memory Benchmark

As there is no temporal database, such as InfluxDB, available on Android, We first compare FLAIR's performances with SQLITE, as it is the only database natively provided on Android.

To compare the memory consumption of the two approaches, two same operations are performed with both SQLITE and FLAIR: *(i)* the incremental insertion of random samples and *(ii)* the incremental insertion of constant samples. The memory footprint on the disk of both solutions is compared when storing timestamped values. As FLAIR models the inserted samples, random values are the worst-case scenario it can face, while inserting constant values represents the ideal one. One million samples are stored and, for every 10,000 insertion, the size of the file associated with the storage solution is saved. The experiments are done with a publicly available application [10].

Figure 6 depicts the memory footprint of both approaches. On the one hand, the size of the SQLITE file grows linearly with the number of inserted samples, no matter the nature (random or constant) of the samples. On the other hand, the FLAIR size grows linearly with random values, while the size is constant for constant values. In particular, for the constant values, the required size is negligible. The difference between vanilla SQLITE and FLAIR is explained by the way the model is stored: while SQLITE optimizes the way the raw data is stored, FLAIR is an in-memory stream storage solution

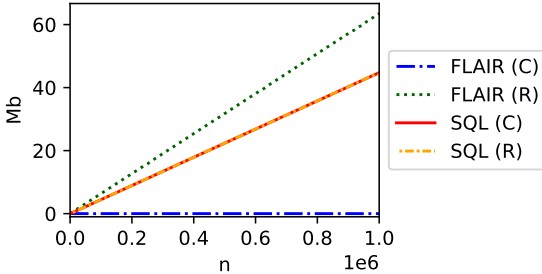

Figure 6: Insertion of 1,000,000 samples, random (R) or constant (C), in both SQLITE and FLAIR.

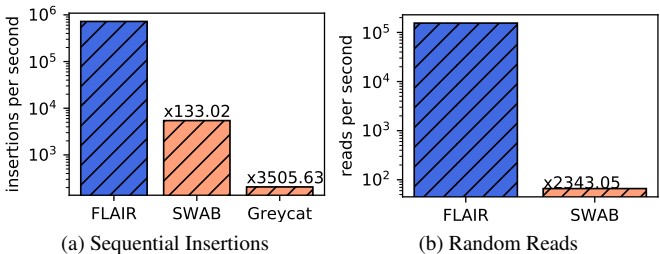

Figure 7: Throughput for insertions and reads using FLAIR, SWAB, and GREYCAT (log scale). FLAIR drastically outperforms its competitors for insertions and reads.

which naively stores coefficients in text file. Using more efficient storage would shrink the difference between the two. As expected, the memory footprint of a data stream storage solution clearly outperforms the one of a vanilla SQLITE database in the case of stable values. While random and constant values are extreme cases, in practice data streams exhibits a behavior between the two scenarios which allows FLAIR to lower the memory required to store those data streams.

In practice, we compare SQLITE and FLAIR to store the entire PRIVAMOV dataset (7.2GB). FLAIR only requires 25MB compared to more than 5GB for SQLITE, despite the naive storage scheme used by FLAIR. On mobile devices, loading the raw dataset in memory crashes the application, while FLAIR fits the same dataset into memory.

## 5.2   Throughput Benchmark

We compare FLAIR with its competitors among the temporal databases: SWAB and GREYCAT. We study the throughput of each approach, in terms of numbers of insertions and readings per second. For the insertions, we successively insert $1M$ random samples in the storage solution (random values are used as a worst-case situation for FLAIR, due to its way of modeling data). For the reads, we also incrementally insert $1M$ samples before querying 10,000 random samples among the inserted ones. GREYCAT is an exception: due to its long insertion time, we only insert 10,000 random values and those values are then queried. Our experiment is done using a publicly available application [13].

Figure 7 shows the throughput of the approaches for sequential insertions and random reads. Note the logarithmic scale. FLAIR drastically outperforms its competitors for the insertions: it provides a speed-up from $\times 133$ against SWAB up to $\times 3,505$ against GREYCAT. The insertion scheme of FLAIR is fast as it relies on few parameters. On the other hand, GREYCAT relies on a costly procedure when a sample is inserted: it tries to increase the degree of the current model until it fits with the new point or until a maximum degree is reached. GREYCAT aims at computing a model as compact as possible, which is not the best choice for fast online inser-

tions. While SWAB performs better, it cannot compare to FLAIR because of the way SWAB inserts a sample: when its sliding window is full and a new sample does not fit the current model, a costly bottom-up approach is triggered over the entire window.

For the reads (Fig 7b), FLAIR also outperforms SWAB. Our investigation reports that the gain reported by FLAIR largely benefits from the time index it exploits to fetch the models: SWAB browses the list of models sequentially until the good model is found while FLAIR relies on a dichotomy search. SWAB has a complexity linear in the size of the models list while FLAIR has a logarithmic one. Nonetheless, their lists of models have roughly the same size as random samples were added. GREYCAT has the same approach as SWAB and this is why it is not represented in the results: with only 10,000 insertions instead of $1M$, its list of models is significantly smaller compared to the others, making the comparison unfair. Nonetheless, we expect GREYCAT to have a better throughput as its model list shall be shorter.

Note that those results have been obtained with the worst-case: random samples. Similarly unfit for FLAIR are periodical signals such as raw audio: our tests show a memory usage similar to random noise. Because FLAIR leverages linear interpolations, it performs best with signals that have a linear shape (e.g. GPS, accelerometer). We expect SWAB to store fewer models than FLAIR thanks to its sliding window, resulting in faster reads. However, the throughput obtained for FLAIR is minimal and FLAIR is an order of magnitude faster than SWAB for insertions, so it does not make a significant difference. We can conclude that FLAIR is the best solution for storing an unbounded stream of samples on mobile devices.

## 5.3   Privacy Benchmark

### 5.3.1   Location privacy

Location data is not only highly sensitive privacy-wise, but also crucial for location-based services. While LPPMs have been developed to protect user locations, they are generally used on the server where the data is aggregated. The data

is thus exposed to classical threats, such as malicious users, man in the middle, or database leaks. To avoid them, the best solution is to keep the data on the device where it is produced, until it is sufficiently obfuscated to be shared with a third-party. With GPS data, this protection mechanism must be undertaken by a device-local LPPM. Evaluating the privacy of the resulting trace must also be performed locally, by executing attacks on the obfuscated data. Both processes require storing all the user mobility traces directly on the mobile. While existing approaches have simulated this approach [25], no real deployment has ever been reported. In this section, we show that using FLAIR enables overcoming one of the memory hurdles of constrained devices. We use FLAIR to store entire GPS traces on mobile devices, execute POI attacks, and protect the traces using the LPPM PROMESSE [35].

PROMESSE [35] is an LPPM that intends to hide POIs from a mobility trace by introducing a negligible spatial error. To do so, PROMESSE smooths the trajectories by replacing the mobility trace with a new one applying a constant speed while keeping the same starting and ending timestamps. The new trace $T'$ is characterized by the distance $\delta$ between two points. First, additional locations are inserted by considering the existing locations one by one in chronological order. If the distance between the last generated location $T'[i]$ and the current one $T[c]$ is below $\delta$, this location is discarded. Otherwise, $T'[i+1]$ is not defined as the current location $T[c]$, but the location between $T'[i]$ and $T[c]$, such that the distance between $T'[i]$ and $T'[i+1]$ is equal to $\delta$. Once all the locations included in the new mobility trace are defined, the timestamps are updated to ensure that the period between the two locations is the same, keeping the timestamps of the first and last locations unchanged. The resulting mobility trace is protected against POI attacks while providing high spatial accuracy.

Our experiments are performed using a publicly available application [7].

**Enforcing privacy on** CABSPOTTING   Using FLAIR, we store the entire CABSPOTTING dataset's latitudes and longitudes in memory, using both $\varepsilon = 10^{-3}$ and $\varepsilon = 2 \times 10^{-3}$ (representing an accuracy of approximately a hundred meters). For each user, we compute the gain in terms of memory we save by modeling the dataset instead of storing the raw traces.

Figure 8 reports the gain distribution as a CDF along with the average gain on the entire dataset. One can observe that most of the user traces benefit from using FLAIR, and FLAIR provides an overall gain of 21% for $\varepsilon = 10^{-3}$ on the entire dataset, and a gain of 47.9% for $\varepsilon = 2 \times 10^{-3}$. Nonetheless, the mobility of a few users imposes an important cost: for them, using FLAIR is counter-productive. Fortunately, this does not balance out the gain for the other users.

To better understand how the $\varepsilon$ parameter introduced by FLAIR affects the utility of the resulting traces, we study the POIs inferred from the modeled traces. We compute the

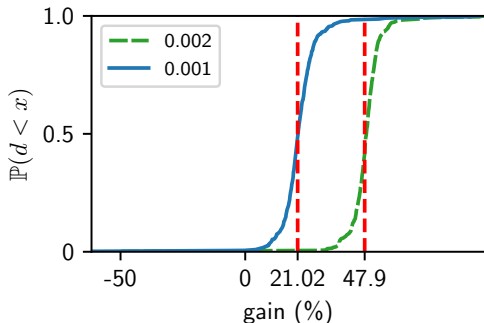

Figure 8: Memory gain distribution when storing CABSPOTTING with FLAIR. Using FLAIR with $\varepsilon = 10^{-3}$ reports on a gain of 21%, while $\varepsilon = 2 \times 10^{-3}$ reaches a gain of 48%.

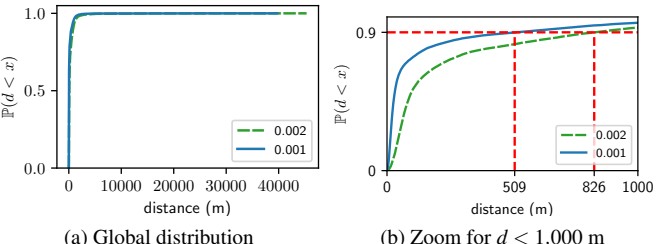

(a) Global distribution              (b) Zoom for $d < 1,000$ m

Figure 9: Distances distribution when using FLAIR on CABSPOTTING. The distances are computed between the POIs obtained using the modeled traces and their closest counterparts, obtained with the raw traces. Except for a few extreme values, the values are close: 90% of the POIs are at a distance lower than 510 meters from the ground truth. The use of FLAIR does not alter the utility of the traces.

POIs of the trace both with and without using FLAIR. To estimate the relevance of the obtained POIs, we compute the distance of each POI reported while using the trace modeled by FLAIR to the closest POI among the POIs in the raw trace. Figure 9 depicts the distribution, as a CDF, of this distance between "modeled" and "raw" POIs. Figure 9a shows that the distance is short: with $\varepsilon = 10^{-3}$, 99.5% of the distances are lower than 2,425 meters and 99% are lower than 1,700 meters. Figure 9b zooms on this distribution, focusing on distances lower than 1,000 meters. With $\varepsilon = 10^{-3}$, 90% of the obtained POIs using FLAIR are at a distance lower than 510 meters to a POI inferred from the raw trace. By construction, POIs are the center of spheres of a diameter of 500 meters where the user has stayed more than 5 minutes. The vast majority of the obtained POIs using FLAIR being within 500 meters of raw POIs, it means that FLAIR delivers relevant approximations. With $\varepsilon = 2 \times 10^{-3}$, 90% of the obtained POIs using FLAIR are at a distance lower than 826 meters to a POI inferred from the raw trace: the gain in memory has an impact on the utility of the resulting trace.

Figure 10 reports on the sensibility analysis of $\varepsilon$, both in terms of gains and distances. As expected, the higher $\varepsilon$, the

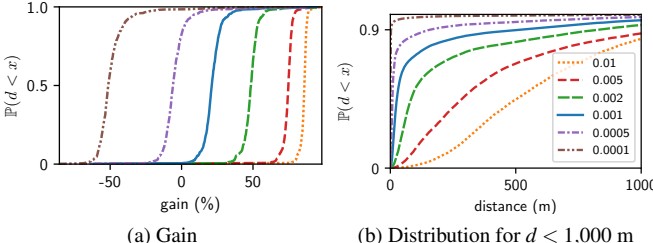

(a) Gain                      (b) Distribution for $d < 1,000$ m

Figure 10: Distances distribution for different ε when using FLAIR on CABSPOTTING. Distances and memory gain are computed from the modeled traces with different values for ε. The higher ε, the higher the gain, but the longer the distances between the inferred and raw POI.

better the gains, but the longest the distances. Regarding the gains (Fig. 10a), a low ε can induce a memory overhead. Indeed, if the model is used only for one data point, it generates a memory overhead similarly to Fig. 6, in this case of 50%. We, therefore, recommend using $\varepsilon = 10^{-3}$ as the minimal tolerated error to observe a gain. Regarding the distances, Figure 10b reports on the distribution of distances below 1,000 meters, as the higher values follow the same tendency as Figure 9a. Except for a few extreme values, most of the distances remain short, even for high ε values.

**Processing Benchmark**  For dense datasets, *e.g.* with more than two GPS samples per second, the gain becomes even more significant. For example, storing the entire PRIVAMOV dataset using FLAIR with $\varepsilon = 10^{-3}$ results in **a memory gain of** 99.87%. Compared to sampling, FLAIR stores all the samples, instead of discarding a part of them. However, the large number of samples can be a hindrance to many approaches, including the extraction of POI. To be able to port LPPMs onto constrained devices, other bottlenecks of the systems should be resolved, in addition to storage.

For example, computing POIs with the traditional POI attacks may lead to unpractical computation time. Computing the POIs of the user 1 of PRIVAMOV takes 2 *hours*: computing the POIs for the entire dataset is far too costly. We cannot expect end-users to execute processes with such computation time on their mobile phone: while FLAIR has removed the memory constraint, computation time is still a hurdle. *Divide & Stay* is a way, in this case, to decrease the complexity of POI computation. Table 1 displays PRIVAMOV user 1 POIs' computation time on different platforms. It shows that applying *Divide & Stay* to the user 1 mobility trace decreases the computation from 2 *hours* to 59 *seconds* on Android, providing a ×120 speed-up; speed gain even reaches ×164 on iOS, computation time decreasing from 1 *hour* to 22 *seconds*. *Divide & Stay* makes the *in-situ* use of POI attacks and the corresponding LPPM possible.

In addition to speed, the quality of the inferred POIs is the most salient concern about *Divide & Stay*. We assess the

Table 1: Computation times of raw POIs for PRIVAMOV user 1 on different platforms. *Divide & Stay* (D&S) is at least 100 times faster than state-of-the-art approaches.

| Platform | POI-attack | D&S | Speed-up |
|---|---|---|---|
| Desktop | 59 min 20 s | 32 s | ×111 |
| iOS | 1 h 00 min 01 s | 22 s | ×164 |
| Android | 1 h 58 min 04 s | 59 s | ×120 |

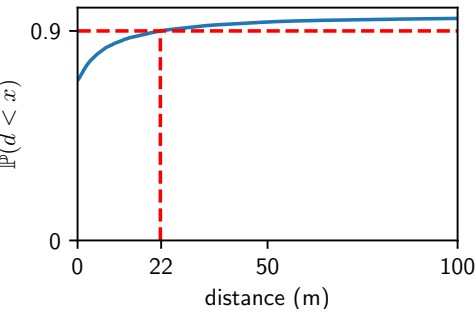

Figure 11: Distances distribution when using *Divide & Stay* on CABSPOTTING. The distances between the POIs are obtained using *Divide & Stay* and their closest counterparts, obtained with the traditional POI attack. Except for a few extreme values, the values are close: more than 68% are the same and 90% of the POIs are at a distance lower than 22 meters than a "real" one.

quality by computing the distances to the POIs obtained from the POI-attack on CABSPOTTING. We choose CABSPOTTING because computing it on PRIVAMOV is prohibitive it terms of computation time. Figure 11 displays the distribution of the distances below 100 meters: more than 68% are the same and 90% of the POIs are at a distance lower than 22 meters from actual ones. *Divide & Stay* provides an important speed-up without altering the quality of POIs. Note that FLAIR was not used in this case, as the performances of *Divide & Stay* are orthogonal to the use of a temporal database to model the samples.

**Bringing back privacy to the user.**  By using both FLAIR and D&S we can perform POI-attacks and use LPPMs directly on the user's device. We consider the POIs of user 0 of CABSPOTTING with and without FLAIR, D&S, and PROMESSE, see Table 2.

The use of FLAIR and D&S alters the number of POIs, which explains the extreme values obtained in the distribution of the distances (Fig. 9 and 10b): it corresponds to POIs that have no counterpart and may be far away from other POIs. The use of D&S corroborates the results of Figure 11: an important part of the inferred POIs look similar to the raw ones. On the other hand, even though the number of POIs is similar, none of the POIs obtained using FLAIR are equal to the original one, with or without D&S, despite being very

Table 2: Impact of FLAIR and D&S on the number of inferred POIs from `user 0` trace in CABSPOTTING. Thanks to FLAIR and D&S, PROMESSE succeeds to protect user privacy at the edge.

| Algorithm | without PROMESSE | | with PROMESSE | |
|---|---|---|---|---|
| | Raw POIs | FLAIR | Raw POIs | FLAIR |
| POI-attack | 30 | 31 | 0 | 0 |
| D&S | 30 | 30 | 0 | 0 |
| POI-attack ∩ D&S | 21 | 20 | - | - |

close.

To conclude, our implementation of the data stream storage solution, FLAIR, enables the effective deployment of more advanced techniques, such as EDEN [25] or HMC [28]. This may require new algorithms, such as *Divide & Stay*, but it enables *in-situ* data privacy protection before sharing any sensitive information. We believe that this is a critical step forward towards improving user privacy as all LPPMs experiments until today were either centralized or simulated.

## 5.4 Stability Benchmark

We further explore the capability of FLAIR to capture stable models that group as many samples as possible for the longest possible durations. Figure 12 reports on the time and the number of samples covered by the models of FLAIR for the CABSPOTTING and PRIVAMOV datasets. One can observe that the stability of FLAIR depends on the density of the considered datasets. While FLAIR only captures at most 4 samples for 90% of the models stored in CABSPOTTING (Fig. 12a), it reaches up to 2,841 samples in the context of PRIVAMOV (Fig. 12c), which samples GPS locations at a higher frequency than CABSPOTTING. This is confirmed by our observations of Figures 12b and 12d, which report a time coverage of 202 ms and 3,602 ms for 90% of FLAIR models in CABSPOTTING and PRIVAMOV, respectively. Given that PRIVAMOV is a larger dataset than CABSPOTTING (7.2 GB vs. 388 MB), one can conclude that FLAIR succeeds to scale with the volume of data to be stored.

## 5.5 Beyond Location Streams

**Storing timestamps** In all the previous experiments, the timestamps were not modeled by FLAIR, as we expect the user to query the time at which she is interested in the samples. However, it is straightforward to store timestamps using FLAIR: we store couples $(i, t_i)$ with $t_i$ being the $i$th inserted timestamp. Unlike other sensor samples, the nature of the timestamps makes them a good candidate for modeling: their value keeps increasing in a relatively periodic fashion. To assess the efficiency of FLAIR for storing timestamps, we stored all the timestamps of the `user 1` of the PRIVAMOV dataset with ε = 1—*i.e.*, we tolerate an error of one second per estimate. The 4,341,716 timestamps were stored using

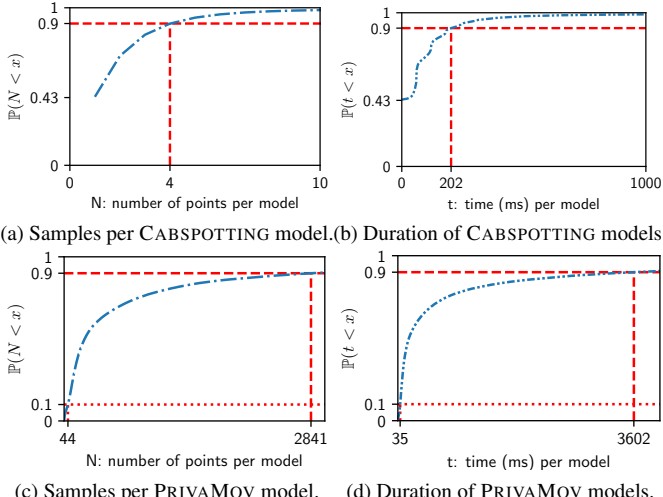

(a) Samples per CABSPOTTING model. (b) Duration of CABSPOTTING models.

(c) Samples per PRIVAMOV model. (d) Duration of PRIVAMOV models.

Figure 12: Stability of the inferred models when using FLAIR on PRIVAMOV and CABSPOTTING with ε = $10^{-3}$.

26,862 models for a total of 80,592 floats and an overall gain of 98%, with a *mean average error* (MAE) of 0.246 second. Hence, not only the use of FLAIR results in a dramatic gain of memory, but it provides very good estimations.

**Storing accelerations** To assess that FLAIR is suitable for storing unbounded data streams, we use FLAIR to store accelerometer samples. While storing random samples is of little benefit, accelerometer samples are used in practice to model user mobility. Coupled with other sensors' data, such as GPS values, we can infer if the user is walking, biking or taking a car for example [19, 38, 41]. However, the accelerometer produces more than 15 samples per second, hence challenging the storage of such a data stream. Our implementation is publicly available [4].

We store 10,000 consecutive accelerometer samples with FLAIR and, for every 100 insertions, we report on the size of the file and the relative gain. We use FLAIR with ε = 1 as the accelerometer has high variability, even when the mobile is stationary. FLAIR reports a constant memory whenever stationary, and a small gain (> ×1.39) when walking. FLAIR is thus a suitable solution to store data streams produced by the sensors of mobile devices.

We also observed that the performances of FLAIR may differ, depending on device configurations. As older hardware's accelerometers are noisier and produce fewer samples than newer sensors, FLAIR's gain appears as higher on latter generation hardware. For instance, inserting 10$k$ samples with a *Pixel 7 Pro (Android 13)* smartphone is completed in 21 seconds, while doing the same on a *Moto Z (Android 8)* lasts for 49 seconds. Regarding iOS, latest *iPhone 14 Plus (iOS 16.0.1)* takes up 1 minute 39 seconds to store same samples count.

## 6  Threats to Validity

While the combination of FLAIR and D&S succeeds to embed LPPMs within mobile devices and increasing user privacy, our results might be threatened by some variables we considered.

The hardware threats relate to the classes of constrained devices we considered. In particular, we focused on the specific case of smartphones, which is the most commonly deployed mobile device in the wild. To limit the bias introduced by a given hardware configuration, we deployed both FLAIR and D&S on both recent Android and iOS smartphones for most of the reported experiments, while we also considered the impact of hardware configurations on the reported performances.

Another potential bias relates to the mobility datasets we considered in the context of this paper. To limit this threat, we evaluated our solutions on two established mobility datasets, CABSPOTTING and PRIVAMOV, which exhibit different characteristics. Yet, we could further explore the impact of these characteristics (sampling frequency, number of participants, duration and scales of the mobility traces). Beyond mobility datasets, we could consider the evaluation of other IoT data streams, such as air quality metrics, to assess the capability of FLAIR to handle a wide diversity of data streams. To mitigate this threat, we reported on the storage of timestamps and accelerations in addition to 2-dimensional locations.

Our implementations of FLAIR and D&S may suffer from software bugs that affect the reported performances. To limit this threat, we make the code of our libraries and applications freely available to encourage the reproducibility of our results and share the implementation decisions we took as part of the current implementation.

Finally, our results might strongly depend on the parameters we pick to evaluate our contributions. While FLAIR performances (gain, memory footprint) vary depending on the value of the $\varepsilon$ parameter, we considered a sensitive analysis of this parameter and we propose a default value $\varepsilon = 10^{-3}$ that delivers a minimum memory gain that limits the modeling error.

## 7  Conclusion

The contributions of this paper are threefold: we introduced *i)* a new storage system based on piece-wise linear model dubbed FLAIR, *ii)* a new way to compute POIs, called *Divide & Stay*, and finally *iii)* demonstrated how FLAIR could unlock device-local privacy protections on time series while using machine learning. Our extensive evaluations, based on real applications available for Android and iOS, show that FLAIR drastically outperforms its competitors in terms of insertion throughput—FLAIR is more than 130 times faster than the traditional SWAB—and read throughput—FLAIR reads 2,340 times faster than SWAB. While FLAIR can store tremendous data on mobile devices, *Divide & Stay* provides an important speed-up to reduce the total computation time of

POI attacks by several orders of magnitude, making them suitable for mobile computing. By sharing these two frameworks with mobile developers, our contribution is an important step forward towards the real deployment of LPPMs and, more generally, privacy-friendly data-intensive workloads at the edge (*e.g.*, federated learning on mobile phones).

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
