# OpenReview forum: "FLAIR: Storing unbounded data streams on mobile devices to unlock user privacy at the edge"
_JSYS/2023/March_Papers — Reject_

### Official Review · Reviewer_BaQf · 2023-03-27

**Decision:**

Weak reject: interesting papers with flaws, not sure if they can be fixed in three months

**Strengths:**

The paper makes an effort in advancing the state of the art for pushing data analysis towards edge devices, which is of utmost importance in modern cyber-physical systems.

The paper has a special focus on privacy, which is also an important motivation behind a better use of the computational power of the cloud-to-edge continuum.

Besides the main contribution (PLA) the paper also develops a complete use case to support its claims.

**Weaknesses:**

The paper does not account for recent work about PLA, edge computing, and stream processing. I believe part of what the authors claim as contributions are actually discussed in such missing references:
- Duvignau et al. 2019. Streaming piecewise linear approximation for efficient data management in edge computing. In Proceedings of the 34th ACM/SIGAPP Symposium on Applied Computing (SAC '19). Association for Computing Machinery, New York, NY, USA, 593–596. https://doi.org/10.1145/3297280.3297552
- Havers et al. DRIVEN: A framework for efficient Data Retrieval and clustering in Vehicular Networks, Future Generation Computer Systems, Volume 107, 2020, Pages 1-17, ISSN 0167-739X, https://doi.org/10.1016/j.future.2020.01.050.

- The discussion provided by the authors about some of their choices (especially in their evaluation, is not discussed/justified)

- The structure of the paper, the order in which aspects are presented, and some missing discussions about the proposed solution result in a paper in which the contribution is not crisply described, with important questions (in my opinion) unanswered.

**Detailed Comments:**

The following is a list of points that I think can help the authors in polishing their work:

1) I recommend the authors check the two papers I mentioned in the weaknesses and discuss them, they will find some of the contributions in them overlap with those of this paper.
2) It personally took me several passes to get to understand the motivation clearly when it comes to the contribution. If my understanding is correct: LPPMs, which rely on techniques such as POIs, need access to large amounts of raw data, and such data is challenging to maintain on a single edge device. Hence, PLA can be used to significantly reduce the data volumes, “adapted” POIs such as D&S can be used over PLA-compressed data, and thus LLPMs can work. If my understanding is correct, I think the authors should make this description clear and crisp. In its current shape, the description is scattered in several parts of the paper and thus not as easy to grasp as it could be.
3) In several parts of the paper, the authors state that PLA “theoretically offers unlimited storage”. I agree that is correct, but in practice, I am not sure what real signal would exist that, being basically constant (if we exclude the case of a too-large-on-purpose \epsilon for PLA) is there in real-world applications. This seems to me a bit of “over-selling” for some corner-case, although the authors might have examples from real-world applications they can provide for their claims.
4) I do not agree (or do not understand, otherwise) the sentence “However, all these techniques require personal sample to be grouped to enforce user privacy”. If one takes a differentially private technique for yes/no voting, for instance, what samples would be grouped in that case?
5) I do not agree (or do not understand, otherwise) the sentence “a storage system based on a fast PLA to store approximate models of any data streams”. First, I do not see why it needs to be fast. More importantly, not all data streams are time series nor carry numerical values, how would one apply PLA to a stream carrying text? I think the paper would improve by having a clear list of assumptions made by the authors about their system and its underlying data.
6) on page 5, there’s a paragraph describing the impact of value \epsilon (just before Algorithm 2). I agree with the discussion, and I acknowledge that, while discussing \epsilon, the authors mention the role of the underlying data too. I would nonetheless raise the discussion about the trade-off being both about \epsilon AND the data. In other words, while a certain value of \epsilon could solve the problem of over-fitting or under-fitting a portion of data, it could be the case that no single value of \epsilon, for an evolving unbounded stream of data is free from the under- or over-fitting problem. That is why I think it is important to stress that this is a challenge that stems from both \epsilon AND the data.
7) when it comes to the datasets used by the authors, I would claim their size can be handled by modern edge devices (especially noting these datasets are aggregated datasets from multiple sources). I understand it is not easy to get hold of real-world large datasets for these types of applications. On the other hand, it is not optimal to have examples that, by being able to run locally run a fully-fledged DB on all the data on a single device go somehow against the original motivation itself. Continuing this point, I also have some doubts about the validity, as a baseline, of a fully-fledged DB versus an ad-hoc implementation of a proposed algorithm
8) In their experiments, the authors chose several values for the \epsilon they use to compress data. Besides the fact that I would always try to put the unit of measure for data that has one, I think it is of key importance to discuss why such values have been chosen. Also, how would in practice a user choose them? Is there some underlying assumption on the values being known? Should a user experiment first to find out the appropriate value? I think these types of discussions are important, especially since the authors stress the need for real-world applicability of proposed techniques.
9) I do not fully understand the linear growth in size when using PLA over “hard-to-compress data”. I was first puzzled by observing a faster-growing trend when comparing FLAIR and SQL, but I thought that could be justified by SQL having some internal optimization for storing its data (figure 6). Nonetheless, the authors later say “Nonetheless, the mobility of a few users imposes an important cost: for them, using FLAIR is counter-productive” which seems to confirm the underlying PLA could result in data inflation. Is that the case? If so, one could in principle transform each pair of successive points (e.g., 2 values) with the offset and slope of a segment (2 values) and thus prevent inflation in the first place. Maybe this is something the authors could clarify.
10) I do not agree (or do not understand, otherwise) the sentence “To avoid them, the best solution is to keep the data on the device where it is produced, until it is...”. Without a formal definition of an attacker model, and an explicit list of assumptions, I think the term “the best” is not a good choice. It can support, sure, but that it is the best is a much stronger statement.
11) In the evaluation, I think it can be beneficial to gather all the information about data and techniques in a single place (e.g., to avoid for instance mentioning sampling suddenly in Processing Benchmark, page 11)

As a final comment, the English production should also be checked. The following is a list of points I found while reading the paper:
- Abstract: which are consumed by machine learning pipelines TO deliver location...
- Introduction: but also producers --> but ARE also producers
- Introduction: I would not classify “a daily routine” as an example of “surrounding environment” but that I guess is up to the authors choice
- Introduction: These data streams... it contains --> they contain
- Introduction: when introducing abbreviations, use capital letters Sensitive Personal Information (SPI) - this appears twice
- Section 2.4: In particular, PLA are --> is
- Section 3.4: the authors write “is achieved by estimating its image” --> I think I understand what the authors mean, but the expression “to estimate an image” could be improved
- There are a couple of colloquial expressions that could be improved “the standard deviation was small”, “tremendous quantity”

**Expertise:**

Actively publishing in this area

**Summary Of Review:**

The paper discusses a compression method, based on Piece-wise Linear Approximation (PLA) that can help maintain large amounts of data within edge devices and, therefore, promote edge-local computations (prior to data sharing) and thus boost privacy.
To show an example of such a privacy boosting benefit, besides presenting their PLA method, the authors also introduce a technique (named Divide & Stay) that can be locally (at the edge) used to identify Points of Interests from the PLA compressed data. Such Points of Interests can then be used by a Location Privacy Protection Mechanism to prevent shared data from disclosing too much information about the user(s) behind such data.

**Useful:**

yes

---

### Official Review · Reviewer_7FLw · 2023-04-09

**Decision:**

Strong reject: this paper has serious problems, fixing it would definitely take more than three months

**Strengths:**

- The paper addresses an important problem of protecting locational privacy while providing utility for locational services.
- The paper presents lightweight algorithms, both compute- and storage-wise, that are suitable for running on mobile devices.

**Weaknesses:**

- The overall presentation of the paper appears misleading in parts. The abstract and introduction describe the technique as a system, but it is rather too simple to be a system. The conclusion states that the presented technique uses machine learning, but this is an overstatement.
- The contribution of the D&S technique appears to be an add-on, rather than an integrated component of the FLAIR technique.
- The evaluation section is not as compelling. The presentation lacks sensitivity testing on a critical design parameter, epsilon, and it is unclear how effective the proposed design protects against POI attacks.

**Detailed Comments:**

I learned a lot from reading this paper, especially on the problems regarding locational privacy. With that said, I can't help but feel that there are important missing pieces in this work.

First, I was excited early on in the paper where it described the work as a storage system. However, I find the main idea of the paper underwhelming, as it does not describe a system. Rather, it is an approximation algorithm for modeling locational data streams as a series of linear vectors. As such, comparing FLAIR to SQLite, a database engine, is not comparing apples to apples. Similarly, the experimental setup and conclusion describe the technique in FLAIR as machine learning. I see this as a bit of a stretch as there is no learning in FLAIR - when the next data point is outside the error margin, it simply resets and starts anew. There is no prediction based on the model or quantifying how the model accurately represents the real dataset. I worry that this comment may seem controversial, and I am not gatekeeping - I am pointing out that I felt misled by the paper.

Second, I did not understand the significance of the D&S technique for the POI attack. I do understand that it would be faster than a full scan (O(logN) vs. O(N)), and that it sacrifices the accuracy of stay in doing so. What I do not understand is, why does computing stay have to be fast or mobile-friendly? Isn't the assumption that the attacker would access the data stream (whether it be FLAIR or the original data points) outside of the mobile device, at the third party? If so, shouldn't the paper assume that the attacker has a much more powerful machine for the attack? On the other hand, if it assumes that the attacker has access to the data on the mobile device, isn't this an entirely different threat model?

Third, the introduction discusses how locational data streams can be used to extract POI or sensitive personal information and that there is an inherent tradeoff between the utility of service and privacy. This is an excellent point but it is unclear how the proposed work addresses these concerns and it is not sufficiently covered in the evaluation. It seems that epsilon is an important parameter that trades accuracy vs. storage space and privacy. Although Figure 5 presents the extreme cases, the paper does not discuss how to arrive at the default 1/100 value. Furthermore, the FLAIR's effectiveness of LPPM is not explicitly shown. Although section 5.3 show that comparison between the original data and the data transformed by FLAIR, what does this mean in terms of the utility of the locational service vs. privacy?

Lastly, I appreciate Figures 6 and 7, on system metrics. I was wondering how FLAIR would perform in worst-case scenarios (random) and I appreciate that the paper discloses this. However, the paper would present a stronger case if it emphasizes that a random locational movement is improbable, and showing the same storage footprint on real traces would be much more helpful. For Figure 7, I am curious to know how much throughput is good enough. More specifically, if only two pairs of GPS samples are produced per second (as explained in the introduction), why is it necessary to achieve close to 1M insertions per second for FLAIR?

**Expertise:**

Follow the literature closely, last published 5+ years ago

**Summary Of Review:**

This paper aims to protect against location privacy attacks (LPA) by implementing linear approximation at the mobile device. The proposed technique, FLAIR (fast linear interpolation), computes and stores a series of linear vectors rather than the measured locational samples, aiming to make it more difficult to extract points of interest (POI) while providing utility for location services. In addition, the paper also presents D&S (divide and stay), a fast and approximate method for computing "stay" that can be used for extracting POI. The presented algorithms are implemented and evaluated on real smartphone devices, covering both Android and iOS.

**Useful:**

no

---

### Official Review · Reviewer_7G5E · 2023-04-17

**Decision:**

Weak accept: good paper with flaws that can be fixed in three months

**Strengths:**

- The authors try to address a real and important challenge, and their approach is rather straightforward.

- The paper contains a thorough experimental evaluation, and the authors also consider potential biases that could occur due to their assumptions.

- Sections 1, 2, and 3 have a good flow, and understanding the proposed technique is rather straightforward.

- The authors provide the source code of the implementation and benchmarks, which can be used to reproduce the results included in the paper or for further evaluation of the proposed technique and implementation.

**Weaknesses:**

- In the paper, "FLAIR" is used to denote both the linear approximation technique and its implementation (presented as a "storage system"). This can be confusing at times, especially when going through the experimental evaluation section.
- The paper could benefit from comparing the proposed approach with more recent work.

**Detailed Comments:**

I have a few questions/comments that I think the authors may consider addressing to enhance the paper:

- Besides addressing privacy-preservation-related challenges, what are the other potential use cases of FLAIR (the data model and/or the implementation)?

- Since the paper presents 1) a linear approximation technique, 2) an implementation of this technique, and 3) an algorithm for POI inference, a figure (or more figures) that indicate the contributions of the paper within a stack of related components/concepts would improve the readability and flow of the paper.
- in section 4.1. - "Memory footprint", "the number of 64-bit variables required by the model" is mentioned as an explored metric, but it does not appear in the current version of the text.

- Section 4.5. can be extended to provide more details about the implementation of FLAIR.

- The third contribution listed in the conclusion can read as if FLAIR uses machine learning, which does not seem to be the case.
- The terms "memory" and "storage" are sometimes interchangeably used, which may confuse the reader. In the same regard, in Section 3.2. it is stated that "... we advocate the use of data modeling... to increase the storage capacity of constrained devices.". This statement can be rewritten to be more accurate, as using FLAIR (or any database/storage system) does not actually increase the storage "capacity" of a device, which is a hardware property.






**Expertise:**

Follow the literature closely, last published 5+ years ago

**Summary Of Review:**

This paper presents 1) a linear approximation technique, 2) an implementation of this technique that can be used to model streaming data points that enables the use of privacy-preserving techniques while maintaining the utility of the data, and 3) an algorithm for POI inference (called Divide & Stay). Through experimental evaluation, the authors argue that their contribution enables practical usage of privacy preservation techniques on resource-constrained and mobile devices, while considering location privacy preservation as the main use case.

**Useful:**

yes

---

### Meta-Review · Area_Chair_MQMK · 2023-04-18

**Recommendation:** Reject
**Confidence:** 5

**Metareview:**

I would like to thank the authors for their submission to JSys. All the reviewers agreed that the paper addresses an important issue of privacy at the edge. Nevertheless, after careful review, I regret to inform you that your submission has been rejected. I recommend that the authors carefully consider the feedback provided by the reviewers to enhance the paper for future submissions. Here are some high-level comments from each reviewer:

* Reviewer 7FLw noted that the paper's abstract and introduction do not accurately reflect its contents, as the presented technique is actually an approximation algorithm rather than a system. Additionally, comparing it with other models, such as SQLite, is deemed invalid.

* Reviewer BaQf pointed out that the novelty of the work is minimal, as the proposed solution has already been presented in other papers not covered in the related work section.

* Reviewer 7G5E found the distinction between the proposed solution being an approximation technique or a system confusing.

---

### Decision · Program_Chairs · 2023-04-20

Reject